# Evaluation of Side Effects and Long-Term Protection of a Sustained-Release Injectable Moxidectin Formulation against *Dirofilaria immitis* Infection in Dogs: An Observational—In Field Multicentric Study

**DOI:** 10.3390/vetsci9080408

**Published:** 2022-08-04

**Authors:** Cristina Vercelli, Luigi Bertolotti, Elisa Gelsi, Carlo Gazza, Giovanni Re

**Affiliations:** 1Department of Veterinary Science, University of Turin, Largo Paolo Braccini 2, 10095 Grugliasco, TO, Italy; 2Azienda Terapeutica Italiana A.T.I. s.r.l., Fatrogroup, Via Emilia, 285, 40064 Ozzano dell’Emilia, BO, Italy

**Keywords:** heartworm, *Dirofilaria immitis*, moxidectin sustained release, dog, safety profile, side effects

## Abstract

**Simple Summary:**

The sustained-release moxidectin formulation of Afilaria SR is labelled to prevent *Dirofilaria immitis* infection in dogs for a period of six months. An observational—in field multicentric study was design to evaluate the tolerability and the long-term prevention of Afilaria SR in Italy. A total of 583 dogs were recruited from 2018 to 2021, receiving the drug annually and monitored by veterinary practitioners after each administration. Antigenic tests were performed 210, 365, 730, and 1095 days after the administration of the drug. None of the enrolled dogs was detected as positive, since it was possible to establish that 100% of protection was achieved. Afilaria SR was well tolerated since only the 13% of dogs demonstrated mild reaction in the injection site and only two dogs out of 583 demonstrated anaphylactoid or angioneurotic reactions. These data support the high prevention rate against *Dirofilaria immitis* disease in all enrolled dogs and indicate the high safety profile of the product, considering the low number and the low grade of side effects.

**Abstract:**

The sustained-release moxidectin formulation Afilaria SR is a relatively new product and has been labelled to prevent *Dirofilaria immitis* infection in dogs for a six months-period. An observational, in field multicentric study was performed, aiming to evaluate the tolerability and the long-term prevention of Afilaria SR in Italy, a country where filariasis is endemic. The study was designed to include not less than 300 dogs, older than 6 months, of any breed. Side effects were recorded by veterinarians and antigenic tests were performed after 210, 365, 730, and 1095 days after the administration of the drug. A total of 583 dogs were recruited from 2018 to 2021 and all of them were negative with respect to antigenic tests at all time points, indicating that 100% of protection was achieved. Ranking of adverse reactions and correlation to patient features were analyzed using descriptive statistics and χ2 square test, respectively. Afilaria SR was well tolerated: 13% of dogs experienced mild reactions and only two dogs out of 583 (0.3%) demonstrated anaphylactoid/angioneurotic reactions, resolved administering corticosteroids. These data support that Afilaria SR prevented *Dirofilaria immitis* disease in all enrolled dogs and the low number and the low grade of side effects indicate the high safety profile of the product.

## 1. Introduction

In the last few decades, the level of care and cure for companion animals has been continuously increasing (especially for dogs and cats). Even if the panorama of therapeutic choices is wide, the best strategy to achieve and maintain an optimal health status is to design a good prevention plan. One of the most important endoparasites in dogs is *Dirofilaria immitis (D. immitis)*, a nematode causative agent of canine heartworm disease characterized by the presence of adult parasites in pulmonary arteries and microfilariae release in the blood stream [1]. *D. immitis* is globally distributed and it is endemic in different temperate regions [2]. Nevertheless, although some differences among countries can be drawn, worldwide the trend of prevalence has been increasing mainly due to climate change and the simultaneous spread of competent mosquitos belonging to the *Aedes* species [3]. In Europe, the situation is varied due to the fact that in some areas, such as Northern Italy, heartworm prevalence is decreasing mainly due to a higher awareness and intense control measures that have been applied in the last few years [4]. Despite this “good news”, filariasis is still considered endemic in Italy [5] and the strict adhesion to preventive plans is mandatory to maintain the situation stable or induce a further decrease [4].

In current practice, the prevention of heartworm disease is performed using macrocyclic lactones (MLs) and several products have been released in the past 30 years, demonstrating both high safety and efficacy profiles [6]. Successful prevention may be compromised by the low compliance of dog-owners with monthly administration, leading to an incomplete and often inadequate protection and, as a consequence, to a final outbreak of heartworm infections [7]. Another concern is related to the possible ML-resistant *D. immitis* spread, which has been already described [1]. In order to reduce the risk of incorrect administration, pharmaceutical companies developed new injectable formulations, mostly based on microspheres, enhancing the sustained-release of the drug in order to achieve a protection lasting for 180 days after a single shot administration [6]. Among the different MLs, moxidectin has been widely studied and nowadays several sustained-release formulations containing this molecule are available all over the world [1].

European Regulation 2019/6 [8] governs the centralized and national marketing authorizations for veterinary medicinal products. This regulation has been in force since 28 January 2022 and repeals the Directive 2001/82/EC and the Regulation (EC) No 726/2004, implemented in Italy by the law 193/2006 [9]. According to these laws, the placement on the market of equivalent veterinary medicine is permitted after an accurate comparison of its efficacy and safety profiles to a previously labeled and commercialized equivalent product present in Italy or in a member State of the European Community, in a period longer than 10 years. The formulation must have the same qualitative and quantitative composition of active substances and the same pharmaceutical form as the reference medicinal product (as well as bioequivalence with the reference medicinal product demonstrated by appropriate bioavailability studies). According to the aforementioned factors, Afilaria SR (ATI s.r.l., Italy) containing moxidectin, was specifically developed and authorized for commercialization in Italy. Similarly, in the case of other drugs based on microsphere technology the release of moxidectin is extended, permitting a single subcutaneous injection that it is expected to protect for the whole vector season against heartworm infection caused by *D. immitis* and to prevent from the infection by *D. repens* [1,10,11]. This formulation is also labelled to protect against diseases caused by *Ancylostomum caninum* and *Uncinaria stenocephala* eventually present at the moment of treatment [1,10].

The safety and efficacy of this treatment have been proven considering the mandatory evaluation procedure prior the commercialization, and it was demonstrated that the administration is possible also in pregnant bitches and in puppies older than 12 weeks [10,12]. Reported side reactions are usually mild and transitory, limited to pain in the injection site, local (i.e., muzzle, paws, eyelids and lips swelling) or generalized (i.e., hives and pruritus) hypersensitivity reactions. Very few intense adverse effects have been reported, such as anaphylaxis, diarrhea, shaking and lethargy. Nevertheless, even if some useful and important information are already available about side effects, the continuing monitoring in the post marketing phase is essential and it is strictly regulated by the veterinary pharmacovigilance European system [8].

According to the aforementioned factors, an observational, in field, multicentric study was designed to enroll owned dogs living in endemic areas of Italy and to evaluate the presentation and the severity of side effects experienced by the dogs receiving Afilaria SR and to monitor in a 3-years period the protection against *D. immitis* infection performing antigenic test prior to and after the administration of the drug.

## 2. Materials and Methods

### 2.1. Study Design

The present observational, in field, multicentric study was designed to enroll at least 300 dogs (sample size estimation given *p* = 0.05, precision d = 0.025 and Z = 1.959 for a 95% confidence level) [13,14], distributed in different regions in Northern-West, Northern-East and center Italy, where filiariasis is endemic: Veneto, Piedmont, Lombardia, Emilia-Romagna, Friuli Venezia Giulia, and Tuscany. Only owned dogs were enrolled, and neither experimental dogs were purchased for scientific purposes nor dogs were experimentally infected. Owner had to sign an informed consent to give permission to the enrollment in the study and to authorize the blood sample collection at different time points.

### 2.2. Dog Enrollment

The inclusion criterion was that the dog must result negative to the IDEXX Snap test 4DX prior the administration of Afilaria SR. a previous treatment with other injectable drugs containing moxidectin or oral drugs containing moxidectin, ivermectin, or milbemycin oxime was not considered an exclusion criterion, whereas a period greater than one year must have elapsed since the last administration in order to exclude possible interference with the present study. Data about the identification of dogs were recorded. Breeds that more frequently show MDR mutation (such as Collies, Borders Collies and Australian Shepherd) have been enrolled, as well.

The dogs receiving the administration of Afilaria SR were included in a vigilance program under the direct supervision of ATI s.r.l. Fatrogroup company. Data about tolerability, efficacy and the results of antigenic tests were independently collected by veterinary practitioners 210, 365, 730, and 1095 days after the administration of the drug (time schedule is carefully described in the paragraph *Heartworm antigenic test)*. In December 2021, at the end of the three-years periods (2018–2020 and 2019–2021), all data were sent to the Department of Veterinary Science of Turin for an independent evaluation by a team composed by two pharmacologists and one biostatistician.

### 2.3. Heartworm Antigenic Test

Whole blood was collected from all dogs at the initial screening visit and during scheduled clinic visits on day 210, 365, 730, and 1095 (±5 days for each visit) after the first administration for the detection of adult *D. immitis* antigen using the commercial kit Snap test 4DX (IDEXX laboratories) following manufacturer’s instructions. The first two time-points were chosen to investigate any latent infections in a period longer than six months, which is the period claimed to be fully protective for the competing product based on sustained-release moxidectin [15]. The following two time-points were performed before renewing the administration, checking the negativity prior offering protection for the following period. Thus, each dog was controlled before the enrollment (T0), and after the treatment at several time points (after 210, 365, 730 and 1095 days).

### 2.4. Drug Administration

Afilaria SR was administered subcutaneously in the interscapular region at the dose of 0.17 mg of moxidectin/kg corresponding to 0.05 mL/kg as reported in SPC indications. The administration was performed annually for the entire observational period (once a year, for three consecutive years), before the high-risk season (from April to September).

### 2.5. Side Effects Reporting and Clinical Evaluation

During both periods of evaluation, dogs were monitored by the veterinary practitioner after the administration of Afilaria SR and by owners at home in order to evaluate the onset of any side effects. Adverse reactions were classified according to the following ranking: 0 = no reaction, 1 = mild, 2 = moderate, and 3 = intense. All reactions were recorded in a specific datasheet, detailing all clinical information, drug administration to resolve the adverse reaction, and outcome of the patient. All information about side effects were signaled to the Italian Ministry of Health according to the current legislation about veterinary pharmacovigilance [8]. The pharmaceutical company was informed and fulfilled its duties specific for manufacturing companies to adhere to the regulations relating to veterinary pharmacovigilance [8].

### 2.6. Statistical Analysis

Data collected were analyzed in order to evaluate the association between the emergence of adverse reactions and animal features. In more detail, the adverse reaction events were recorded for each period, classified for severity and associated to animal breed, age (younger or older than four years old) and the time of the first administration. The association between adverse reaction frequencies and animal features was evaluated using χ^2^ test. The part concerning clinical observation required only descriptive statistics.

## 3. Results

### 3.1. Animals

A total of 583 dogs have been enrolled in the present study, corresponding to the sample size needed to evaluate a prevalence of 0.05 with a 99% confidence level, improving what was initially designed (*n* = 300). A previous treatment with injectable moxidectin formulation was received by 304 dogs while ivermectin or milbemycin tablets were administered to 164 dogs. For all of them, it was possible to assess that these administrations were performed a year before the enrollment in the present study, thus respecting the precautionary wash-out period that was preliminary assessed to avoid any bias or interference. The remaining part of the enrolled dogs (*n* = 115) received Afilaria SR as first treatment to prevent Dirofilaria immitis infection.

All dogs tested negative with respect to the antigenic test performed prior to the first administration of Afilaria SR.

No mortalities have been recorded among dogs involved in this study related to the administration of Afilaria SR. More detailed information is provided in the following sections.

### 3.2. 1st Period: 2018–2020

In the first period of vigilance, 418 dogs were enrolled: they were mostly mix-breed (274 out of 418) and distributed in Lombardia (*n* = 147), Piedmont (*n* = 75), Veneto (*n* = 69), Tuscany (*n* = 61), Emilia Romagna (*n* = 54) and Friuli (*n* = 12).

In this period, the majority of side effects were recorded in 2018 (Figure 1a); few mild adverse reactions were recorded. It was demonstrated that a significative correlation existed between the young age of animals (less than four-years-old) and reporting side effects (χ^2^
*p* < 0.05). It was not possible to assess a statistically significant association between breed or first administration and the occurrence of adverse effects. In 2019 and 2020, it was not possible to delineate a correlation between the onset of side effects and any of the considered parameters. Among the dogs presenting side effects, 13 animals that demonstrated a mild reaction in 2018 did not demonstrate any type of reactions in the following years. Seven dogs demonstrated a mild side effect in 2018 and in 2020 while the rest did not show anything in 2019. Six dogs presenting a mild reaction in 2018 demonstrated a milder reaction in 2019 (Figure 2a).

### 3.3. 2nd Period: 2019–2021

In the second period of vigilance, 165 dogs were enrolled belonging to Lombardia (*n* = 60), Veneto (*n* = 49), Romagna Emilia (*n* = 56). Also, in this case, mix-breed dogs were prevalent (118 out of 165).

Also, during the second period, the younger dogs (less than four-years-old) receiving for the first time the administration of the drug in 2019 demonstrated a statistically significant correspondence to the occurrence of side effects (χ^2^ *p* < 0.05). During 2020 and 2021, no correlations have been drawn between the onset of side effects and the considered parameters. Among the dogs presenting side effects, only one dog presented an intense reaction in 2020 after having shown a moderate reaction in 2019 and without a relapse in 2021. Another dog showed an intense reaction in 2019 and nothing more in 2020. Six dogs presenting a mild reaction in 2019 did not show any reaction after the administration in the following years. Seven dogs presenting mild reactions in 2019 and 2020 did not present anything in 2021 (Figure 2b).

### 3.4. Evaluation of Side Effects

As previously mentioned, the side effects were ranked according to the intensity of signs and symptoms shown by dogs.

In the first period, 53 dogs presented adverse reactions after the administration of Afilaria SR. A total of 46 out of 53 demonstrated swelling, mild pain, and pruritus in the injection site for 24 h following the administration, while 3 out of 53 demonstrated the same symptoms for 48 h and 2 out of 53 for 12 h.

Only one dog demonstrated weakness for 24 h after the treatment while another dog showed moderate facial swelling and diffuse presence of hives: in the latter case, the veterinarian decided to administer dexamethasone to limit the symptoms and the patients’ outcome was excellent in 24h and it was not necessary to exclude it from the study.

In the second period, 23 dogs experienced side effects: 12 dogs demonstrated mild pain at the injection site for 24 h after the drug administration (5 out of 23 for 12 h and 3 out of 23 for 48 h). Only one dog demonstrated weakness for 12 h after the treatment with Afilaria SR and in another dog was described paleness of the mucous membranes the day after the administration. In this period, only one dog demonstrated a moderate adverse effect showing an angioneurotic facial edema characterized by mild swelling of maxilla and periorbital region. No life-threatening symptoms appeared, and the recovery of the dog was complete after the administration of dexamethasone and fluids.

### 3.5. Heartworm Antigenic Test

All enrolled dogs were screened 210, 365, 730 and 1095 days after the administration of Afilaria SR and all resulted negative.

## 4. Discussion

The present observational, in field, multicentric study was designed to study enrolled owned dogs living in endemic areas of Italy, aiming to evaluate the onset and the severity of side effects to Afilaria SR administration and the long-term protection derived from a regularly scheduled prevention administering Afilaria SR.

The initial design aimed to enroll 300 dogs but the compliance of the owners and of the veterinary practitioners was enormous, leading to a final recruitment of 583 dogs. The large number of enrolled dogs treated with the same formulation was higher than any other recently reported study performed in clinical conditions to evaluate the safety and efficacy profile of moxidectin sustained-release formulation [6,16,17]. One might object that no placebo group was designed in the present study, which might be considered a limiting factor of the present study. The authors would like to underline that this is an observational in-field study based on the vigilance of onset of possible adverse effects in a post marketing panorama. A placebo group is requested in pilot studies and in pre-marketing phase, when a brand-new product is under investigation prior commercialization. In this specific case, Afilaria SR is already commercially available, labelled for the prevention of *D. immitis* in dogs. According to the fact that all legal permissions and authorizations have already been granted, the use of a placebo group was not mandatory and, considering that only owned dogs in endemic areas have been enrolled, administering a placebo should be seen as an unethical issue.

The efficacy of moxidectin sustained-release injectable formulation has been recognized since the beginning of this decade and was defined as to complete protect dogs against *D. immitis* for twelve months even in experimental infection conditions [6,18,19]. Moxidectin is more lipophilic in nature than ivermectin and other avermectins, can easily undergo to redistribution, and has a long half-life that makes it suitable for long-acting formulations [1,20]. Moreover, the pharmaceutical techniques allow one to achieve a longer release period, potentiating the intrinsic pharmacokinetic properties of the molecule. The sustained delivered system circumvents owners’ compliance issues relating to the monthly administration of other preventive drugs, such as topical macrocyclic lactones [21].

The study of Krautmann et al. [16] stated that the administration of moxidectin as a sustained-release formulation did not induce side effects. This is also in accordance with the study of Heaney and Lindahl [21], where the safety profile of injectable formulation of sustained release moxidectin was evaluated in 10 weeks old puppies treated with 3× and 5× the normal dose. No adverse effects were observed in puppies treated 3× and only mild swelling in the injection site was reported due to residual microspheres in the group that received a 5× dose. In these dogs, mild depression of erythropoiesis was also observed, even if all parameters remained in normal limits without any symptoms. No physical or neurological alterations have been recorded [21].

Comparing the results obtained in the present study with those obtained in the observational clinical study of McTier et al. [6] most of the dogs recruited experienced at least one side effect following the administration of ProHeart 12 or Heartgard Plus (87.9% and 85.1%, respectively). The most common side effects were vomiting, lethargy, diarrhea, and anorexia that affected dogs the day after the administration of both drugs. Mild injection site reactions occurred in few dogs treated with ProHeart and resolved spontaneously in seven days. Only 2% of dogs treated with both drugs experienced an anaphylactoid/hypersensitivity reaction. Comparing these data to those obtained in our study, the incidence of side effects induced by the administration of Afilaria SR is lower than that assigned to the other formulations labelled for prevention of *D. immitis* disease considered in the study of McTier and colleagues [6]. In fact, among the 583 dogs enrolled, only one dog experienced an anaphylactoid reaction and only another dog showed angioneurotic symptoms, corresponding to an incidence of 0.34% of all enrolled dogs.

The results of this study indicate that all enrolled dogs resulted negative for *D. immitis* at all time points of 210,365, 730, and 1095 days after the drug administration. These data is important since it supports the notion that the prevention induced by a regular administration of Afilaria SR permit one to achieve long protection. Afilaria SR and competing products are claimed to prevent the infection in the high-risk season, which in Italy lasts from April to September. In the present study, Afilaria SR was administered prior the high-risk season in order to achieve the maximum prevention and accordingly to the summary of product characteristics (SPC) indication. Considering that all enrolled dogs were annually checked prior the new administration for three consecutive years and that all dogs resulted negative to antigenic test, it is reasonable to hypothesize that Afilaria SR is responsible for a protection longer than six months. These data are in accordance to what was already demonstrated by Lok et al. [15]: dogs experimentally exposed to *D. immitis* were treated with Guardian SR (the first moxidectin injectable sustained released formulation labelled to prevent *D. immitis* infection in dogs) reaching a protection period of 12 months. This should be seriously taken into consideration, considering the fact that climate changing is responsible of a long-lasting high-risk season, and vector can live longer [4]. Nevertheless, prevention against filariasis must be regularly scheduled and administered. Only through a complete awareness of practitioners and owners about the adherence to preventive treatments could the decreasing trend of filariasis in Italy, as described by Genchi and Kramer [4], be maintained. The reason behind this condition is mainly attributable to the strong adhesion to preventive plans performed in the past years resulting in more concrete protection. Comparing these data with recent efficacy evaluation of other marketed formulations labelled to prevent *D. immitis* disease, it appears that Afilaria SR has a higher capacity to prevent the infection of *D. immitis* in the high-risk season and in the post treatment period than Heartgard Plus that has a 1.8% of dogs resulting positive at the antigenic test 365 days administration while Afilaria SR demonstrated 100% of efficacy as well as ProHeart 12 [6].

The screening was performed using SNAP assay (IDEXX laboratories), a commercially available test, that was demonstrated to be the best performing with 100% specificity of and 94.1% of sensitivity [1,22]. For this reason, this test was chosen for the present investigation. The European Scientific Counsel Companion Animal Parasites (ESCCAP) [5] guideline established that diagnosis of filariasis can be performed using Knott test or antigenic test, while European Society of Dirofilariosis and Angiostrongilosis (ESDA) [23] recommend that both tests must be performed in order to diagnose the disease. The present observational study relied on the high performances of the SNAP assay to monitor the prevention of filariasis in dogs enrolled. The lacking execution of Knott test might be considered a minor limitation of the present study, but the Authors are confident that the obtained results are reliable due to the fact that the test were regularly performed for the entire periods of observation (three years).

Considering that no dogs tested positive during the entire observation period, it was established that it was not necessary to investigate possible mechanisms of resistance that are currently under investigation in other countries and are considered to be responsible of the increasing rate of diagnosis of *D. immitis* in dogs [24]. The authors are confident about the soundness of the results obtained in the present study considering the long period dedicated to the observation of the patients (from 2018 to 2021), the huge number of patients that have been enrolled, and the fact that it was included the period from middle May to middle November, which was predicted to be a crucial transmission time of *D. immitis* in Spain, Italy, and Greece [4].

The pharmacovigilance report of possible side effects might be extremely frequent during the first year after the marketing of a new drug: with an unfamiliar molecule, practitioners are more likely to report adverse reactions. Once they become familiar with the drug, they tend not to report subsequent reactions [16]. Also taking, advantage of this phenomenon, the authors are confident to have obtained coherent reports of adverse effects. It is important to underline that the actual European Regulatory [8] maintained the former scope of veterinary pharmacovigilance and improved the serval field of application. The necessity to maintain a high level of vigilance in the post marketing phase of all veterinary drugs make it essential to continuously collect new information about possible side effects for the patients or for the person handling the animal, understand which drug associations are avoided or allowed, or evaluate decreased efficacy [8]. The only method to collect a huge amount of information is to hasten the communication among the veterinary practitioners and their owners in order to be updated about every possible side reaction following a drug administration. Pharmacovigilance signalment has become easy, rapid, and eco-friendly in the last two years in Italy due to the integrated function in the electronic national prescription system. Using the app from mobile phones or a web interfaces veterinarians can send directly and for free their signalments to the Italian Ministry of Health, thus guaranteeing an immediate communication of a suspected adverse reaction [25].

## 5. Conclusions

Filariasis continues to be an important parasitic pathology affecting dogs all over the world, and prevention is the most important way to ensure the maintenance of good health status. In the present study it was demonstrated that a regular prophylaxis plan implemented by administering Afilaria SR, sustained-release formulation of moxidectin provides protection against *D. immitis* infection for a long period considering that all dogs resulted negative to all of the antigen tests performed during the entire observational period. Moreover, the formulation is well tolerated, with only few dogs demonstrating mild reactions at the injection site and an extremely low incidence of anaphylactic reaction. These data are encouraging and could be useful to increase the compliance of owners to administer this drug in order to reach a consistent protection against *D. immitis* infection.

## Figures and Tables

**Figure 1 vetsci-09-00408-f001:**
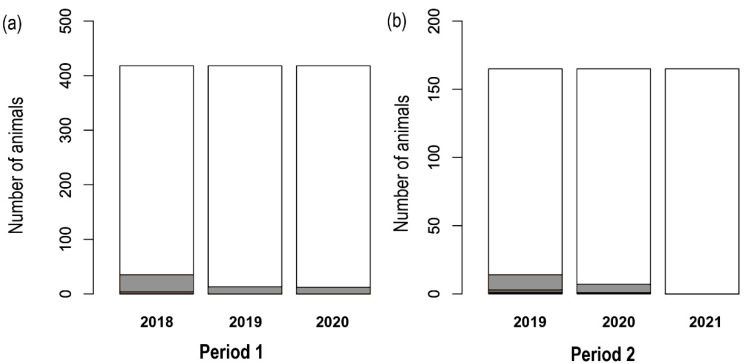
Distribution of adverse reactions during the first (**a**) and the second (**b**) study period. Severity of the reactions is reported in different colors (black: severe; dark gray: moderate; light gray: mild; white: no reaction).

**Figure 2 vetsci-09-00408-f002:**
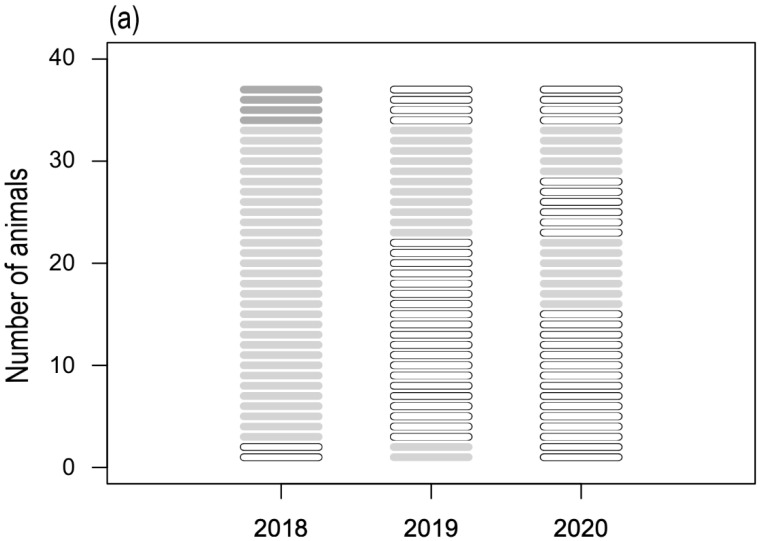
Trend of adverse reactions recorded in the first year of enrollment and monitored during the first (**a**) and the second (**b**) study period. Severity of the reactions is reported in different colors (black: intense; dark gray: moderate; light gray: mild; white: no reaction). Each row represents the same animal tested during the study period.

## Data Availability

Not applicable.

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
