# Peer review of "Evaluation of Side Effects and Long-Term Protection of a Sustained-Release Injectable Moxidectin Formulation against Dirofilaria immitis Infection in Dogs: An Observational—In Field Multicentric Study"

_vetsci, 2022, doi:10.3390/vetsci9080408_

Round 1

Reviewer 1 Report

The presented study shows an interesting information about the Afilaria SR, the anti-parasitic drug with a sustained-release injectable moxidectin to prevent Dirofilaria immitis infections in dogs.

Aim of the study is properly formulated. I strongly recommend shortening and paraphrasing the title of the article, because is too complex.

A simple summary and an abstract sound almost identical - please make the simple summary less detailed.

Figure 1 (a and b) - Please reduce the values on axes

Corrections:

Line 63 - leading to (...) leading to (...) - maybe: "and as a conequence to.."

Line 67 - sustained-release

Line 127 - 210, 365, 730 (...)

Line 170-171 - infromatins -> informations

Line 347 - an extremely (...) anaphylactic

Please add the additional information about the past anti-parasitic prevention of dogs analysed in the study. Whether the use of other injectable drugs with moxidectin was not able to influence on the results?
How many dogs have administered the moxidectin in the past?

References - position 9 ??- please expand the abbreviation

Author Response

Reviewer 1

  • The presented study shows an interesting information about the Afilaria SR, the anti-parasitic drug with a sustained-release injectable moxidectin to prevent Dirofilaria immitis infections in dogs. à Thank you for the time that you dedicated to our manuscript. All comments and suggestions have been taken into consideration. Here a point-to-point response.
  • Aim of the study is properly formulated. I strongly recommend shortening and paraphrasing the title of the article, because is too complex. à Thank you for the suggestion. Our purpose was to highlight the most important features of the manuscript in the title but we agree that it was too long and too complicated. We rephrased it in order to be easier to understand.
  • A simple summary and an abstract sound almost identical - please make the simple summary less detailed. à Thank you for the suggestion. We simplified and rephrased the simple-summary.
  • Figure 1 (a and b) - Please reduce the values on axes à the values have been reduced.
  • Corrections:
    • Line 63 - leading to (...) leading to (...) - maybe: "and as a conequence to.." à fixed
    • Line 67 - sustained-release à fixed
    • Line 127 - 210365, 730 (...) à fixed
    • Line 170-171 - infromatins-> informations  à fixed
    • Line 347 - anextremely (...) anaphylactic à fixed
  • Please add the additional information about the past anti-parasitic prevention of dogs analysed in the study. Whether the use of other injectable drugs with moxidectin was not able to influence on the results? How many dogs have administered the moxidectin in the past? à Thank you for your comments that gave us the opportunity to better discuss in L176-182 what was mentioned in L116-118 about previous treatments.
  • References - position 9 ??- please expand the abbreviation à The reference is about a confidential document of the company that has been better detailed. In the present version the reference is #10

Reviewer 2 Report

This is a study evaluating the efficacy of a commercial Dirofilaria immitis prevention drug in client own dogs in Italy. The results indicated that no Dirofilaria immitis infection was detected in any dogs during the 3 year observation period and only a few mild adverse reactions were recorded. The authors concluded that the commercial drug's efficacy to protect client owned dogs from Dirofilaria immitis is proved and is well tolerated. 

The field study design have some limitation, such as the exposure of dogs to  heartworm is not guaranteed or controlled. One can argue that the result of negative heartworm antigen test during the observation period is due to no exposure to heartworm rather than protection from the prevention agent. I think the authors should discuss about this limitation more clearly. Maybe provide prevalence data of each sample region will help reader to have a better imagination of heartworm exposure to dogs in this study. 

Specific comments:

Line 139 - Dose it means that Afilaria SR is injected once a year in this study? I can't find a clear statement about how Afilaria SR was administered after the first injection. Please clarify.

Line 286 - this hypothesis confused me. If Afilaria SR is administered 6 months apart as label, the hypothesized extension of protection is not validated. If Afilaria SR is administered 12 months apart, it will be an off-label use and should be clearly informed in the manuscript. Also, this study is not designed for evaluating the extension protective period of Afilaria SR, so results can't support the hypothesis. In a word, I think this hypothesis should be deleted.

Figure 1 - Description of the vertical axis in the line chart is missing. Is it number of dogs? Please revised.

Figure 2- Description of the vertical axis in the line chart is missing. Is it number of dogs? There were 37 dogs in 2a and 15 dogs in 2b, but I can't find the same number of dog with adverse effect in the manuscript. In line 212 and line 220 showed different number of dogs with adverse effect. please clarify. 

Conflict of Interest - I notice that 2 authors (E.G. and C.G.) is affiliated with ATI s.r.l., Italy. Are they employee of ATI? If yes, I think it should be indicated in the conflict of interest. 

Author Response

  • This is a study evaluating the efficacy of a commercial Dirofilaria immitis prevention drug in client own dogs in Italy. The results indicated that no Dirofilaria immitis infection was detected in any dogs during the 3 year observation period and only a few mild adverse reactions were recorded. The authors concluded that the commercial drug's efficacy to protect client owned dogs from Dirofilaria immitisis proved and is well tolerated. . à Thank you for the time that you dedicated to our manuscript. Here a point-to-point response.
  • The field study design have some limitation, such as the exposure of dogs to  heartworm is not guaranteed or controlled. One can argue that the result of negative heartworm antigen test during the observation period is due to no exposure to heartworm rather than protection from the prevention agent. I think the authors should discuss about this limitation more clearly. Maybe provide prevalence data of each sample region will help reader to have a better imagination of heartworm exposure to dogs in this study. àDogs lived in an endemic area at the moment of the enrollment and for the entire duration of the observational study. According to ESCCAP guidelines, Italy is considered an endemic country for both D. immitis and D. repens. We add the reference in the introduction to clarify the point. The administration of Afilaria SR has been performed to protect dogs in the high-risk season, that in Italy corresponds to April to September. This was already stated in different parts along the text, and we rephrase some points in order to clarify.
  • Specific comments:
    • Line 139 - Dose it means that Afilaria SR is injected once a year in this study? I can't find a clear statement about how Afilaria SR was administered after the first injection. Please clarify. à Thank you for your comment that gave us the opportunity to clarify. Please, check L150-152.
    • Line 286 - this hypothesis confused me. If Afilaria SR is administered 6 months apart as label, the hypothesized extension of protection is not validated. If Afilaria SR is administered 12 months apart, it will be an off-label use and should be clearly informed in the manuscript. Also, this study is not designed for evaluating the extension protective period of Afilaria SR, so results can't support the hypothesis. In a word, I think this hypothesis should be deleted.: à We did not perform an off-label use. The drug was administered accordingly to the SPC, thus prior the high-risk season, in order to prevent D. immitis infection in dogs for 180days. We did not change what SPC says, we just observe the long term effects concluding that after a single (annual) drug administration before the high risk season (when dogs have an higher possibility to be exposed to vectors), with dogs continuously living in endemic areas, a negativity to antigenic tests performed once a year for three consecutive years was achieved (each test was performed prior to the “annual” Afilaria SR administration). Thus, it is reasonable to suppose that in the aforementioned conditions, the sole reason of that negativity is that Afilaria SR can effectively prevent D. immitis infection for a period longer than 6 months. We explained all the aforementioned topics rephrasing sentences in L302-313.
    • Figure 1 - Description of the vertical axis in the line chartis missing. Is it number of dogs? Please revisedà fixed
    • Figure 2- Description of the vertical axis in the line chartis missing. Is it number of dogs? There were 37 dogs in 2a and 15 dogs in 2b, but I can't find the same number of dog with adverse effect in the manuscript. In line 212 and line 220 showed different number of dogs with adverse effect. please clarify.  à Figures 2a and 2b are representative of the trend of adverse reactions recorded in the first year of enrollment and monitored for the entire periods of observation. We modified the caption of Fig.2 and b  in order to clarify.

  • Conflict of Interest - I notice that 2 authors (E.G. and C.G.) is affiliated with ATI s.r.l., Italy. Are they employee of ATI? If yes, I think it should be indicated in the conflict of interest. àTwo authors are affiliated with ATI Fatro group but no conflict of interest exists since the data were independently analyzed and evaluated. We already stated the involvement of each author in the present study to the Editorial Office and no conflict of interest has been recognized. We declare our transparency: we are ready to send all the drafts where it is proven that the two authors affiliated to ATI Fatro group just provided grammar or typos corrections and did not influence the evaluation performed by academic authors. Academic authors did not receive money and will not receive any salary after the publication by ATI. To prove or transparency, we signed the “conflict of interest form” as requested by the Editorial Office confirming what was already reported in the manuscript.

Reviewer 3 Report

The manuscript submitted by cristina et al was aimed to investigate the long-term effectiveness of Moxidectin formulation against Diroflaria immitis infection in owned dogs. The major limitation of this study was the absence of placebo control. How authors can justify that this formulation is preventive to D. immitis infection if they don’t compare with the infection in placebo group. Therefore, my opinion is against the publication of this study.

Author Response

Reviewer 3

The manuscript submitted by cristina et al was aimed to investigate the long-term effectiveness of Moxidectin formulation against Diroflaria immitis infection in owned dogs. The major limitation of this study was the absence of placebo control. How authors can justify that this formulation is preventive to D. immitis infection if they don’t compare with the infection in placebo group. Therefore, my opinion is against the publication of this study.

Dear Reviewer 3, here a point-to-point response:

  • The manuscript is by Vercelli et al. (not Cristina).
  • Afilaria SR is already registered to prevent filariasis for a period of 180 days. It is not a drug underwent to registration procedure. According to this, a placebo group is not mandatory. The point is about D.immitis but also (and moreover) a matter of pharmacovigilance and equivalent products (already registered!). We are sure about this due to the fact that one author (Prof. Re) is diplomate ECVPT and another (Dr. Vercelli) is ECVPT resident, eligible. If you have any question about the registration procedure, feel free to detail your doubt to Prof. Re, who sat for long time in EMA and is in charge as consultant of Italian Ministry of Health and of Italian Federation of Veterinary Chambers about legislation of veterinary medicines. For any doubt about pharmacovigilance, Dr. Vercelli is responsible for the Regional Center of Veterinary Pharmacovigilance of Piedmont.
  • We already explained in the original version of the manuscript that a placebo group was not mandatory due to the fact that the drug is already licensed and labelled and that considering the fact that this was not an experimental exposure to D. immitis but an observational study during normal clinical practice
  • Ethical issue: it is not ethical to think about the fact that owned dogs remain exposed to D. immitis having the possibility to treat them with a registered drug.

Round 2

Reviewer 3 Report

Thanks for addressing the issues. I have no further reservation.